# Alternative between Revitalisation of City Centres and the Rising Costs of Extensive Land Use from a Polish Perspective

Aleksandra Jadach-Sepioło [1,2,*] and Maciej Zathey [3]

1 Institute of Urban and Regional Development, 03-728 Warszawa, Poland
2 Warsaw School of Economics, 02-554 Warszawa, Poland
3 Institute for Territorial Development, J. Wl. Dawida 1a, 50-527 Wroclaw, Poland; maciej.zathey@irt.wroc.pl
* Correspondence: ajadach-sepiolo@irmir.pl

**Abstract:** The phenomenon of spatial chaos is ever-growing challenge in Poland. Its most common explanations are the weaknesses of spatial planning and the households' economic-based decisions of building a house in the suburbs. In this context, Polish publications lack analyses of the impact of local authorities' on shaping conditions for the development of new housing and renovation of the existing ones. The authors put forward a thesis about the persistence of an extensive land use policy model in Poland, in which local governments create conditions favouring area-consuming approach to locating buildings. At the same time, the same local governments allow de-agriculturalisation of land plots with a consequence that newly developed areas are not equipped with utilities (e.g., sewage or heating networks). Chaos in the development of residential areas is also illustrated by another phenomenon. Local authorities designate large degraded and revitalisation areas. This results in the dispersed effects. The article concentrates on these three symptoms of spatial chaos in Poland, i.e., random and dispersed expansion of new investments in sewage system, lack of integration between district heating systems and direction of residential development and dispersed effects of revitalisation, which cannot prevent flight from blight. The obtained results allowed to confirm the thesis about the extensive land use policy model in Poland.

**Keywords:** spatial chaos; extensive land use model; land use policy; revitalisation; revitalisation areas; sewage network; heating network; GHG emissions

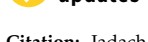



## 1. Introduction

Increasing concerns about sustainability of urban development have resulted in a growing number of policies, governance, and management models oriented at encouraging land-consumption behaviours. They are an ever-growing subject of scientific research, in which the milestones were set years ago by the works of Alonso [1], Muth [2,3], Mills [4], Fujita [5,6], or Krugman [6]. Many other researchers explain detailed mechanisms referring to the context of specific countries [7–14]. Alonso [1] added a spatial dimension to the analysis of households' consumption choices. He defined an ideal city inhabited by rational households, and jobs concentrated in the central district. The household maximise their utility by an optimal set of choices referring to the quality of residential location, quantity of residential space, distance from the city centre and quantity of other consumption goods. The optimal set of consumption goods (including the best residential location) cannot exceed the budget constraint for every rational household. A spatial equilibrium exists when no household has any incentive to relocate or to alter its consumption basket [1,15,16]. The model of spatial equilibrium explaining households' residential locations was extended by adding new inhabitants. This extension enabled to analyse the impact of the new households' residential decisions on the well-being of the previous inhabitants in the locations they considered optimal in the sense of distance to the city centre and set of consumption goods in the particular location [17,18]. In addition to economic models, significant explanations are provided by hedonic analysis framework pioneered by Rosen [19]. This research

framework examines the impact of the attributes of a house, the locational attributes and neighbourhood, city, and region's quality on the home prices. Improvements in a house's external characteristics are likely to increase its price by reducing blight in neighbourhood, providing better employment, shopping, culture, and leisure opportunities, and improving the image of a particular location [20–25].

The population growth and suburbanisation pressure occurred in many countries in Western Europe and North America after World War Second with strong negative effects. Gradually, the planning systems of the majority of these countries began to react to the implications of suburban sprawl, depending on the adopted model of national spatial policy [10–14,26,27]. Hence, another important research framework was established with main focus on land use planning and its impact on the future land use patterns. The first approach to the management of land use change is restrictive, introduced by developing guidelines and documents (especially local plans) based on the principles of sustainable development. The second one, adaptive and liberal, starts with the theory of social participation and incorporates methods of public consultations to achieve consensus of different interest groups on the future functions of a particular land [28]. The choice of approach is rather an evolutionary process than a one-off decision, and in each country it derives from many factors. Among them, the key ones comprise demographic changes (population growth), expectations on the standards of living, as well as the national model of spatial policy that results in the specific set of tools used to reduce sprawl [29–33]. Detailed research on the effectiveness of anti-sprawl models was conducted by Bento, Franco and Kaffine, analysing commonly used instruments, such as development taxes, urban growth boundaries, property taxes, and gasoline taxes [34]. Their analyses prove that the effectiveness of the anti-sprawl inventory is a derivative of the location of land (urban city core, the urban suburbs, and rural area). Contemporary research shows that national urban plans have the potential to foster sustainable land-use at the national scale but have limited power within cities. However, their results depend on the formation of local coalitions to limit city boundaries in local plans which may prove difficult in many municipalities. Even if national governments try to reduce urban sprawl, stakeholders at the local level are in fact interested in expanding urban land use [35].

Research on the symptoms and costs of spatial chaos has been one of the most important research trends in Poland in recent years [36–38]. So far, the most cross-sectional and synthetic approach was presented by Śleszyński et al. [39]. In the Polish context, the most relevant causes of urban sprawl result from households' decisions on their location (microeconomic approach emphasised in by Lityński, Hołuj [40]) and weaknesses of spatial policy in the country at all administration levels [39,40]. Śleszyński et al. [39] provide the main cost categories of spatial chaos, listing economic costs, strictly social costs, environmental costs, and public utility costs, understood as external effects of development dispersion (higher financial expenses for the development of infrastructure and delivery of public services). The estimated annual cost of spatial chaos is, according to Śleszyński et al. [39], EUR 19.92 billion, including the following costs:

— building infrastructure and providing public services to dispersed recipients;
— prolonged commuting to work, congestion, waste of time during travels between the place of work and home;
— exclusion of good soils from agricultural production, reduction of forests and green areas;
— compensation related to the expansion of the transport system, limited revenues from property taxes;
— increasing expenditure on nature and health protection; natural disaster damage removal.

The overall estimation of the costs resulting from spatial chaos is difficult mainly due to the lack of a direct relationship between the expenses incurred and urban sprawl. Most of the costs result from external effects, difficult to express in money, which makes measurement even more difficult [39,41,42].

An additional difficulty is the discrepancy in the perception of these costs by a specific household, local economy of a suburban commune (Polish: gmina) and a city. From the

Polish household's perspective, the cost of building new house in a suburban commune is significantly lower than buying an apartment in a city centre, due to the much lower cost of a building plot. The whole investment cost in suburbia does not include the deferred cost of commuting, as it is balanced by subjectively highly rated benefits: the proximity to nature (own garden, and fresh air) and privacy (contact with nature within the family circle). From the local economy point of view, in a suburban commune the environmental costs of urban sprawl [29–32] do not outweigh the benefits, because the inflow of new residents (especially well-to-do middle class representatives) translates into the increase of tax revenues. Changes in the natural environment or gradual decrease of agricultural land are too delayed to be highlighted, discussed, and internalised. The population growth is the basic goal and the object of competition between neighbouring communes. From this perspective, the direct costs of urban sprawl (e.g., ineffectively large and dispersed water and sewage investments, new roads, etc.) are positively perceived by suburban authorities as pro-development expenditure. These investments also meet the expectations of residents who opt for the improvement of transport accessibility and better access to the water, sewage, and gas networks [35,39]. O'Sullivan points out some of the most relevant market failures which result in incomplete information about the costs of individual housing investment in suburban real estate market:

− external costs of congestion (including air pollution, time-consuming commuting to work and school, and stress) are not included in the cost of purchasing building plot in the suburbs,
− subsidising flats or houses bought/rented on the primary market from public funds with a rather low permissible price per square metre, stimulating the purchase or rental of real estate in cheap locations, mainly on the outskirts of cities,
− cost of building technical infrastructure and social facilities is not taken into account in the cost of purchasing building plots in the suburbs,
− cost of developing vacant land and the loss of landscape values or free biologically active space not taken into account in the costs of purchasing building plot in the suburbs [43].

When explaining the causes of urban sprawl, most authors in Poland emphasise at least one of abovementioned market imperfections referring to failure to internalise the costs of:

− new dispersed infrastructure investment in the suburban locations in the building plot value, as well as the investment process, and living in suburbia,
− commuting to the city centre to job, school, shopping, or entertainment [44,45].

According to Pigou's theory of welfare, although households' location decisions to move to suburbia increase the individual utility, their negative impact on the community is overwhelming [17,18,44,45]. In addition to environmental and social costs, new investments in dispersed infrastructure for sprawled outskirts is a classic case of market failure, as a group of people use infrastructure, for which all city inhabitants pay [46]. Introducing an additional fee to enable internalisation of these hidden costs seems to be fair solution [47].

Some scientists indicate a close relationship between the city's degradation or blight and urban sprawl [48]. "Flight from blight" is one of the main reasons why urban containment boundary fails as a policy [49]. Mieszkowski and Mills described the theory of flight from blight as natural for individual households desiring to avoid any manifestation of social and economic blight, and infrastructure, etc. The most important factors are crime, poverty, and poor housing conditions [48]. Bento, Franco and Kaffine emphasise that there are two contradictory processes determining the scale of urban sprawl. Namely, the low cost of individual housing investment in the suburbs and the downtown deterioration [50].

Urban sprawl in Poland is usually explained by the weaknesses of spatial planning and huge discrepancy between land value in suburban areas and downtowns. In fact, urban sprawl in Poland does not differ much from similar processes in the Mediterranean

countries [51]. The latter explanation may be thus helpful in better understanding the Polish phenomenon. Despite the initially high-density development, a tendency towards sprawl has been visible in the last thirty years in Mediterranean cities (especially in Italy). In many cases, the process was organic and chaotic. It was not restricted by planning strategies but left to market forces driven by the ever-growing migration of inhabitants to suburbia caused by crises in cities' centres. The economic recession in southern Europe has significantly affected competitiveness and weakened the economic performance of entire cities. An example of a key to the interpretation of the forms of urban sprawl in Italy is the concept of a "crisis city" presented in the study of Naples [52]. The urban sprawl intensified after Naples had become a crisis city. Chaotic degradation of the city due to the gradual weakening of planning control in attractive locations and the decline of poorer districts became symptoms of a crisis path lock-in with no possibility to move forward towards mature urban models [53]. This process led to depopulation, pauperisation and criminalisation of some Naples' neighbourhoods, and, at the same time, to the loss of valuable agricultural and recreational land on the outskirts of the city. Although in Polish conditions, crisis phenomena in cities do not occur on the scale of Mediterranean cities, many small and medium towns are in recession due to the collapse of industry, because the process of social and economic transformation started in the 1990s has created a similar shrinking city context [53].

The evidence from Italian cities provide arguments linking the Polish model of sprawl with deteriorated city centres [54,55]. On the other hand, some Scandinavian cities give some hope that the revitalisation processes in Polish cities will mitigate sprawl. Copenhagen, currently assessed as the most attractive and friendly to residents, begin to recover only in the mid-1990s. Before, in the 1980s and early 1990s, there was a shortage of jobs, while pauperisation and crime were visible symptoms of the crisis [56]. Financially, the city of Copenhagen was close to bankruptcy at the turn of the 1980s and 1990s [57]. The city and its urban region have been revitalised. Today they are a strong national centre of economic growth [58], and the Ørestad district is one of the most important European examples of the importance of environmental issues in urban renewal [59].

The problem of biodiversity loss and increasing GHG emissions in suburban communes, as well as peripheral buildings in urban commune are becoming an increasingly important for the processes of urban sprawl in Europe and all over the world [60]. A specific kind of cost of spatial chaos is the energy consumption of the settlement structure, which is a derivative of weakness of spatial planning. This problem, from the theoretical point of view, has been noticed in the source literature since the 1970s energy crisis [61–67]. It should be noted, however, that some theoretical studies on the settlement structure are in close connection with the minimisation of energy supply costs in the US in the first half of the 20th century [68]. One of the first studies that systematically took into account the cost of dispersal of buildings was the report "The Costs of Sprawl" [69]. Despite this long-lasting theoretical discussion, the practical application is still lacking [70–73], what is also raised and shown on examples of extensive spatial development in this article.

In reference to the above-mentioned discussion in the source literature, the main goal of the article is to present the coexistence of two contradictory trends in the spatial management and policy in Poland:

— consent of local authorities to urban sprawl, proved by the scale of dispersed investment in new infrastructure (especially sewage systems) in newly developed suburban areas;
— lack of concentration of revitalisation activities in deteriorated city centres proven by the extensive areas of intervention.

As a consequence, a phenomenon that intensifies spatial chaos in Polish cities can be observed. Namely, increasingly housing sprawl is accompanied by technical degradation of central areas, especially in small and medium-sized towns. Designating extensive revitalisation areas does not favour their effective renewal, but results in the dispersion

of investments over a large area, so the effects are less tangible. Municipal authorities compete for inhabitants by choosing an extensive model of space management:

— Firstly, they create conditions for locating buildings on the largest possible area of the commune. At the same time, they do not impose restrictions on the development of areas that are not equipped with amenities (e.g., sewage system or district heating network). As a result, new housing developments, regardless of heating network routes, increase $CO_2$ emissions and air pollution in a vast area of the commune.

— Secondly, local authorities try to attain external subsidies for revitalisation activities, so they set the intervention area as vast as possible. As a result, they obtain dispersed effects which, by contrast, further supports "flight from blight" attitude of the residents.

In order to verify the abovementioned, the following specific objectives were formulated:

— presentation of the scale of changes in equipping communes with technical infrastructure on the example of a sewage network;

— presentation of the Polish approach to designating extensive revitalisation areas with numerous inhabitants;

— presenting the discrepancy between the district heating network routes in communes and the directions of building developments in them.

The analysis for the first two specific objectives was carried out at the level of the entire country, while for the third one, the results of the research carried out by the Institute for Territorial Development (ITD) for the Dolnośląskie Voivodeship (Province) were used.

The choice of two types of utilities (sewage and heating) resulted from the specificity of the expansion of these two kinds of the networks in Polish communes. Access to water supply is already treated as a basic standard in most communes, while investments in the development of sewage networks come later. Therefore, the analysis of the increase in the network length, especially in urban–rural and rural communes, where there was no significant increase of population, provides an illustration of the ineffectiveness of these investments with a low indicator of the number of inhabitants per 1 km of the network [38,39]. The discrepancy between the directions of the development of the heating network and the directions of development of residential areas are another example of inefficiency or even spatial chaos. The environmental effects of this process make the problem not only important from a scientific but also a practical point of view.

Due to the multithreaded nature of the study, the introduction showed a broad context of the analysed tendencies of urban sprawl in Polish and in foreign source literature. Then, the method of analysis was presented, with its respective stages corresponding to respective specific objectives. The part of the article devoted to the results focused on the discussion of both contradictory directions of an extensive spatial management and policy model in Poland.

## 2. Research Methods and Materials

For the purpose of illustrating the coexistence of two abovementioned contradictory trends in the spatial management and policy in Poland the following analyses were carried out:

— statistical one, i.e.,

  ○ scale of investment in new infrastructure on the example of the sewage network was identified,

  ○ the surface and the number of inhabitants of the degraded and revitalisation areas,

  ○ coexistence of the dispersed sewage systems and designation of extensive degraded and revitalisation areas,

— spatial and geographic in the GIS framework—spatial dependence between directions of urban development and the location of new investment in district heating systems (DHS) in the Dolnośląskie Voivodship.

First, the data available in Statistics Poland (Polish: Główny Urząd Statystyczny, GUS) was used to identify changes in length of the sewage network. For each commune in Poland, according to the administrative division in 2019, the increase in the length of the sewage network in 1995–2019 was calculated. Then, the results were aggregated to the regional level with the total values for following types of commune:

- urban, including the cities with county rights (Polish: miasta na prawach powiatu);
- urban–rural;
- rural (The division into urban, urban–rural, and rural communes (Polish: gmina) is conventional-administrative and not functional (i.e., the urban–rural type in any case does not indicate a transition between typical urban and rural areas)).

In the next step, the data from the survey "Statistical data on revitalisation at the level of the communes" (carried out by Statistics Poland as a part of the project "Statistics for cohesion policy. Support for the monitoring system of the cohesion policy in the financial perspective 2014–2020 as well as programming and monitoring of the cohesion policy after 2020"). A full study was carried out, covering 2478 communes, in two editions (data for 2015–2016 and 2017) [74] were analysed. In particular, the surface of the degraded and revitalisation areas was analysed in the communes where revitalisation programmes were in force at the end of 2018. A total of 2469 communes responded to the survey, so only 9 communes in Poland did not submit the completed questionnaire. From this perspective, the collected data is excellent empirical material for analysing dependencies on a national scale. The latest available data was used.

According to the data of Statistics Poland, at the end of 2018, the total number of revitalisation programmes in Poland was 1494, including 1167 revitalisation programmes developed on the basis of the Act on Municipal Self-Government (hereinafter: the Act on Municipal Self-Government [75]) (PR/LPR—simplified document) and 327 communal revitalisation programmes on the basis of the Act on Revitalisation [76] (GPR—full document).

Degraded areas in communes with the PR/LPR are on average three times larger than the areas of revitalisation, while in communes with GPR—almost five times larger. In terms of the population, both degraded and revitalisation areas in communes with GPR were almost twice as densely populated. Revitalisation areas were, in turn, three times more densely populated than degraded areas (regardless of the type of programme). That shows that the revitalisation intervention focuses on a much smaller area than the area identified by the communes as degraded (but still large) and covers as many of its inhabitants as possible. The average share of the revitalisation area in a commune varies between 3.2 and 5.8% depending on the type of commune, while the share of the number of inhabitants of the revitalisation area in the population of the commune oscillates around 20%. Therefore, the concentration of revitalisation intervention is not the strength of Polish communes.

To verify the main thesis of the article, the causality was not analysed, but the coexistence of phenomena (the expansion of the sewage network and the designation of vast degraded and revitalisation areas) with the Spearman's rank correlation coefficients [77]. The Spearman's coefficient is more universal than the Pearson's correlation coefficient which measures only a linear relationship because it allows to determine the strength of a monotonic co-relation, which may be non-linear.

It is assumed that for $R_i$ denoting the observation $x_i$ and $S_i$ denoting the observation $y_i$, $\overline{R}$ and $\overline{S}$ denoting the mean values of the respective $R_i$ and $S_i$. The Spearman's coefficient is expressed by the following relationship:

$$r_s = \frac{\sum_{i=1}^{n} \left( R_i - \overline{R} \right) \left( S_i - \overline{S} \right)}{\sqrt{\sum_{i=1}^{n} \left( R_i - \overline{R} \right)^2} \sqrt{\sum_{i=1}^{n} \left( S_i - \overline{S} \right)^2}} \tag{1}$$

The variable X denotes the increase in the length of the sewage network in the respective communes, while the variable $Y_1$—the surface of the degraded areas in the communes, $Y_2$—the surface of the revitalisation areas, $Y_3$—the population of the revitalisation areas.

After obtaining the results on the relationship between the variables X and $Y_1$ as well as X and $Y_2$ or X and $Y_3$, tests for statistical significance were performed using the Student's t-test. For statistically significant correlation coefficients, the type and strength of correlation were examined, assuming that:

−  | r_s | < 0.2 is no dependency,
−  0.2 ≤ | r_s | < 0.4 is weak correlation,
−  0.4 ≤ | r_s | < 0.7 is moderate dependency,
−  0.7 ≤ | r_s | < 0.9 is strong dependence,
−  | r_s | ≥ 0.9 is very strong dependence.

A positive, statistically significant relationship means that the analysed phenomena coexist in the communes. Namely, in those which note an increase in the sewage network as well as extensive degraded and revitalisation areas. The analysis was carried out for all 16 voivodeships for all types of the communes. The division by the types of the commune was necessary because the results obtained at the voivodeship level turned out to be statistically significant for all types of communes only in case of the relationship between the variables X and $Y_3$, but only for the urban communes in the case of the relationship between the variables X and $Y_1$ as well as X and $Y_2$.

Research of this kind has not yet been published. Scholars active in the field of urban studies did not refer to the coexistence of these particular symptoms of spatial chaos at the national level. Maybe it is only the case of Poland, but the results may also be interesting for the European perspective, because the scientific literature does not contain analyses for all revitalisation areas at a national level. Validation of the relationship between the approach to the expansion of the sewage network and the designation of a vast revitalisation area may show an additional specificity of spatial chaos in Poland. Urban sprawl causes ineffective expenses for the dispersed sewage infrastructure, which cannot be internalised in the fees from the owners of neighbouring properties. The study made it possible to check whether this ineffectiveness is accompanied by other, namely—ineffectiveness in conducting revitalisation process on too large area.

The spatial and geographic analysis of a spatial dependence between directions of urban development and the location of new investment in district heating systems (DHS) in the Dolnośląskie Voivodeship was carried out on the basis of data from the Institute for Territorial Development (ITD) in Wrocław [78]. The study was limited to the cities with DHS managed by companies licensed to conduct business activities in the field of heat generation, transmission and trade, the list of which is published in the form of a register by the Energy Regulatory Office (ERO, Polish: Urząd Regulacji Energetyki).

The concessions were granted by the ERO in the following 30 cities in this voivodeship: Bogatynia, Bolesławiec, Brzeg Dolny, Chocianów, Chojnów, Dzierżoniów, Głogów, Jawor, Jelcz Laskowice, Jelenia Góra, Kamienna Góra, Kłodzko, Legnica, Lubań, Lubin, Milicz, Nowa Ruda, Oleśnica, Oława, Polkowice, Ścinawa, Siechnice, Stronie Śląskie, Świdnica, Świebodzice, Wałbrzych, Wrocław, Ząbkowice Śląskie, Zgorzelec, and Złotoryja. Data shared by the respective heating companies was the basis for the analysis. Data on the district heating network and buildings connected to the heating system were used to designate areas with access to the heating network, and to perform calculations.

The examination of the spatial dependence between directions of urban development and the location of new investment in the DHS based on the content of the Studies of the conditions and directions of the spatial development (Study, Polish: Studium uwarunkowań i kierunków zagospodarowania przestrzennego), which define the local land use policy.

The analyses were carried out on the basis of the Studies of the conditions and directions of spatial development for the end of 2017, because at this time the anti-smog resolutions was adopted by the Dolnośląskie Voivodeship parliament. Due to the heterogeneous structure of the prepared documents, a generalisation was made and a coherent method of determining the area indented use was adopted. The analyses took into account the areas indicated in the Study drawings as residential, service, and residential and service buildings, or mixed function areas with a predominant residential function. The selected

areas were generalised as housing and service areas (HSA). Such approach also resulted from the analyses carried out by the Provincial Inspectorate for Environmental Protection in Wrocław, which had stated that the emission of pollutants into the air, in particular the emission of suspended dust PM 10, PM 2.5, and B (a) P, comes mostly from housing and service areas.

Based on the data from the analysis of planning documents and data obtained from district heating network operators, the ranges of the heating network in respective communes in relation to service and housing areas were determined. To determine the range of the network, a 200-metre buffer around the buildings connected or planned to be connected to the heating network was adopted. Buildings connected (or planned to be connected) to the heating system were identified on the basis of the layer of existing buildings, obtained from the Database of Topographic Objects for 2015 and supplemented with the objects from the Open Street Map or local geoportals.

The full scope of the analysis and the list of housing and service areas are included in the report [78]. The examples presented in this article were selected because of the largest size of housing and service areas (HSAs) that also reflects an extensive approach of local land use policy.

### 3. Results

#### 3.1. Dispersed Sewage Networks in Poland

The relationship between the chaotic urban sprawl and the development of technical infrastructure is the subject of many analyses in the Polish source literature, despite the emphasised methodological difficulties [79–82]. Along with the ever-growing tendency to urban sprawl, land use policy in Poland is becoming more and more extensive [28]. Low coverage with local spatial development plans and the communal authorities' competition for residents mean that more and more areas are being developed, especially as residential ones. According to research by P. Śleszyński [81], some areas in Poland note even a peak in the supply of construction land because of de-agriculturalisation of land plots. These areas are usually equipped with no technical infrastructure. Along with their dynamic development, however, there is growing social pressure on local authorities to equip newly built housing areas with a sewage network, etc. Śleszyński [81] additionally emphasises that because of many years of negligence regarding the extension of the sewage networks, the ineffectiveness of its development in dispersed settlement systems became a particular problem. The length of the network per capita is many times higher in the areas with a dispersed settlement structure. The scale of investment in new infrastructure in a sewage network was examined on the basis of the data from Statistics Poland. The obtained results are summarised in the Table below:

The problem of excessive expenditure for construction and maintenance infrastructure is well identified in the literature, especially in rural areas in Poland with a very strong dispersion of settlements [82,83]. Infrastructural underinvestment and ineffectiveness of various types of networks has a strong negative impact on the environment and landscape. According to the data of Statistics Poland, between 2016 and 2019 the number of inhabitants connected to the water supply network raised from 84.1 to 92.2%, and to the sewage system—from 49.1 to 71.2%. A significant part of the investment presented in the Table 1, especially in rural areas, has taken place in this period. As a result of scattered and chaotic land use, the networks are longer than optimal and inefficient. This is illustrated by the network length indices per inhabitant, many times higher in areas with an dispersed settlement structure. In some communes this indicator even exceeds 100 metres for one person [82].

Table 1. Increases in the length of the sewage network in the Polish communes in 1995–2019.

| Voivodeship | Number of Communes Analysed | Increase in the Length of the SEWAGE Network (km) | In Urban Communes (km) | In Urban–Rural Communes (km) | In Rural Communes (km) |
|---|---|---|---|---|---|
| Dolnośląskie | 169 | 8614.2 | 1532.2 | 2642.1 | 4439.9 |
| Kujawsko-Pomorskie | 144 | 6502.4 | 1347.5 | 1490.0 | 3664.9 |
| Lubelskie | 213 | 5421.6 | 1169.4 | 675.9 | 3576.3 |
| Lubuskie | 82 | 3492.5 | 715.8 | 1620.7 | 1156.0 |
| Łódzkie | 177 | 5215.4 | 1469.1 | 1182.5 | 2563.8 |
| Małopolskie | 182 | 14,172.0 | 1821.8 | 4640.1 | 7710.1 |
| Mazowieckie | 314 | 14,749.8 | 5059.0 | 2989.3 | 6701.5 |
| Opolskie | 71 | 4408.9 | 574.4 | 1996.5 | 1838.0 |
| Podkarpackie | 160 | 16,179.6 | 1860.9 | 4122.0 | 10,196.7 |
| Podlaskie | 118 | 2950.6 | 763.0 | 731.8 | 1455.8 |
| Pomorskie | 123 | 8888.6 | 1467.6 | 1404.0 | 6017.0 |
| Śląskie | 167 | 12,736.2 | 6160.8 | 1077.2 | 5498.2 |
| Świętokrzyskie | 102 | 5497.3 | 529.8 | 2164.4 | 2803.1 |
| Warmińsko-Mazurskie | 116 | 5800.5 | 556.3 | 2131.3 | 3112.9 |
| Wielkopolskie | 226 | 12,379.0 | 1588.8 | 5552.0 | 5238.2 |
| Zachodniopomorskie | 113 | 5867.7 | 810.5 | 2548.3 | 2508.9 |

In the context of a very high increase of sewage network length, the low participation of the owners of neighbouring properties in the financing new infrastructure should be noted [84,85]. The adjacent fee which should ensure this participation is reluctantly set and enforced by the municipal executive authorities. There are several reasons for that. One of the most important is the freedom in determining the percentage rate of the adjacent fee in the municipal council resolutions, foreseen in the Real Estate Management Act [86]. Pursuant to Art. 146 para. 2, the fee cannot exceed 50% of the difference between the value of the property before the construction of technical infrastructure devices and the value of the property after the construction. Setting an upper limit without specifying a minimum amount results in a moral hazard. Municipal councils, succumbing to social pressure, set the percentage rate at a low level, e.g., 5% or 10%. As a result, the mayor usually withdraws from determining the amount of the fee, stating that the received income from a particular property would not cover the cost of the appraisal study necessary to determine a specific increase in the value of the property. Another important reason is that establishing an adjacent fee for a specific property is optional. Pursuant to Art. 146 para. 2, it is obligatory for the municipal council to determine the percentage rate of the adjacent fee by resolution, but the amount of the specific fee may be determined each time by the mayor's decision within three years after creating the conditions for connecting the property to individual technical infrastructure devices or after creating conditions for using the constructed road.

The Polish authors emphasise methodological difficulties concerning the identification of funds allocated to the development of technical infrastructure in the areas of scattered housing development in municipal capital expenditure [39,82]. The analysis and verification of the dispersion of sewage networks on the basis of data available in public statistics is no less difficult. For the purposes of the article, it was assumed that the increases in the length of the sewage network in the period present in public statistics (1995–2019) will illustrate, at least approximately, these trends, if they are examined for the entire country. The analysis covered all communes in Poland divided by the type of local government unit and voivodeship (Table 1).

For the purposes of the article, data reflecting the increase in the length of the sewage network in the respective types of the communes according to percentiles were analysed (Table 2).

**Table 2.** Percentiles and the average increase in the length of the sewage network in the respective types of the communes [km].

| Percentiles | Cities with County Rights [1] | Urban Communes [2] | Urban–Rural Communes [3] | Rural Communes [4] |
|---|---|---|---|---|
| 1. | 55.47 | 11.30 | 9.55 | 2.80 |
| 2. | 76.36 | 19.26 | 16.70 | 9.80 |
| 3. | 88.94 | 24.93 | 23.30 | 16.30 |
| 4. | 116.66 | 31.60 | 30.00 | 23.72 |
| 5./median | 153.75 | 42.20 | 39.00 | 30.70 |
| 6. | 204.94 | 48.94 | 53.80 | 41.68 |
| 7. | 248.12 | 62.15 | 70.05 | 54.60 |
| 8. | 304.48 | 78.80 | 93.00 | 72.04 |
| 9. | 447.16 | 103.67 | 128.25 | 106.86 |
| 10. | 251.80 | 207.90 | 435.30 | 373.10 |
| Average: | 238.48 | 51.11 | 58.13 | 46.61 |

[1] $n = 66$; [2] $n = 236$ (without cities with county rights); [3] $n = 638$; [4] $n = 1537$.

The presented results indicate that, regardless of the type of the commune, in 60% of the cases the increase is lower than the average for the group. This indicates a significant discrepancy in expanding the sewage system in each group. In order to compare the scale of this phenomenon, the relation of the average value for each group and the value for the first percentile in a group as well as the relation of the highest value in a group to the value for the first percentile were calculated (Table 3).

**Table 3.** Differentiation of the increase in the length of the sewage network in the respective types of the communes.

| Description | Cities with County Rights | Urban Communes | Urban–Rural Communes | Rural Communes |
|---|---|---|---|---|
| Average/lowest percentile | 4.30 | 4.52 | 6.09 | 16.65 |
| Highest percentile/lowest percentile | 45.28 | 18.40 | 45.58 | 133.25 |

The greatest disproportions were observed in the rural communes. In the case of the urban–rural communes and cities with county rights, a similar differentiation in the increase in the length of the sewage network was noticed, however, with almost six times higher values of the increase for the cities with county rights. The lowest internal differentiation was observed in the urban communes. The discussed results show that the expansion of the sewage network in the scattered building development is strongly observed in the urban-rural communes and in the rural communes located in the functional areas of large cities, as well as in the cities with county rights. The expansion of the sewage network in these communes is accompanied by stagnation in the case of urban–rural and rural communes (60% of all types of the communes) located peripherally, where the extension of the network is too small to catch up with the years of underinvestment.

*3.2. Extensive Degraded Areas and Revitalisation Areas*

In accordance with Art. 2 para. 1 of the Revitalisation Act, revitalisation is a process of recovering degraded areas from the crisis state, carried out in a comprehensive manner through integrated activities for the benefit of the local community, space and economy, territorially concentrated, carried out by revitalisation stakeholders on the basis of the municipal revitalisation programme (GPR) [76]. According to this definition, this process should be territorially concentrated to enable mitigation of negative phenomena.

In accordance with the Revitalisation Act, the first stage of the revitalisation process is a delimitation of the spatial concentration of negative phenomena in the communes, i.e., a degraded area. Art. 9 para. 1 of the Act contains its definition, according to which it is an area in crisis due to the concentration of negative social phenomena, in particular unemployment, poverty, crime, low education or low social capital, as well as insufficient

level of participation in public and cultural life [ . . . ] where, in addition, there is one or more of the following negative phenomena: economic [ . . . ], or environmental [ . . . ], or spatial and functional [ . . . ], or technical. The diagnosis should be developed for the entire commune, using objective, measurable and verifiable indicators. The reference level for determining the occurrence of negative phenomena should be the average noted for the commune. The occurrence of crisis phenomena in the social sphere is a precondition for a given area to be considered a degraded area. The degraded area delimitation should be justified by quantitative data, supplemented, if necessary, by the qualitative studies [87].

According to the data, at the end of 2018, degraded areas covered 3.74 million hectares, i.e., 11.9% of the Polish territory. The total surface and population of degraded areas as well as revitalisation areas are presented by the voivodeships in the Table below (Table 4):

**Table 4.** Surface and population of the degraded and revitalisation areas in the Voivodeships.

| Voivodeship | Number of Communes Analysed | Total Surface of the Degraded Areas (ha) | Total Population of Degraded Areas (pers.) | Total Surface of the Revitalisation Areas (ha) | Total Population of the Revitalisation Areas (pers.) |
|---|---|---|---|---|---|
| Dolnośląskie | 122 | 184.34 | 751.23 | 62.02 | 522.33 |
| Kujawsko-Pomorskie | 125 | 279.69 | 470.53 | 157.27 | 357.79 |
| Lubelskie | 132 | 462.74 | 489.19 | 170.33 | 334.51 |
| Lubuskie | 49 | 119.19 | 233.21 | 35.49 | 191.02 |
| Łódzkie | 53 | 39.18 | 439.22 | 21.44 | 339.23 |
| Małopolskie | 147 | 300.73 | 946.66 | 48.58 | 454.62 |
| Mazowieckie | 154 | 359.76 | 842.53 | 117.43 | 651.52 |
| Opolskie | 32 | 30.26 | 217.16 | 7.66 | 155.08 |
| Podkarpackie | 98 | 291.65 | 571.61 | 69.61 | 325.20 |
| Podlaskie | 55 | 206.24 | 312.40 | 53.63 | 216.61 |
| Pomorskie | 32 | 36.82 | 256.54 | 3.49 | 165.48 |
| Śląskie | 123 | 162.66 | 1156.80 | 57.02 | 897.86 |
| Świętokrzyskie | 87 | 267.13 | 386.17 | 77.54 | 292.62 |
| Warmińsko-Mazurskie | 42 | 42.91 | 279.66 | 13.07 | 250.34 |
| Wielkopolskie | 149 | 492.36 | 1.004.67 | 140.27 | 595.83 |
| Zachodniopomorskie | 94 | 468.37 | 570.32 | 123.68 | 332.57 |
| Total: | 1494 | 3744.00 | 8927.88 | 1158.52 | 6082.61 |

The analysis of these areas by the type of local government unit shows that the largest share of such area in the entire area of a commune was recorded in the cities with county rights [88]. The total surface of degraded areas in the respective types of the communes is presented in the Table below (Table 5):

**Table 5.** Surface of degraded areas in the respective types of the communes.

| Type of the Commune | Total Surface of Degraded Areas (ha) | Share of Degraded Areas in the Area of a Commune (%) |
|---|---|---|
| Urban commune | 196.15 | 13.83% |
| including cities with county rights | 107.43 | 14.38% |
| Urban–rural commune | 1479.71 | 13.92% |
| Rural commune | 2068.13 | 10.76% |
| Total: | 3744.00 | 11.99% |

For the purpose of the article, data on the size of the degraded areas by the types of the communes according to percentiles were analysed (Table 6).

**Table 6.** Percentiles and the average size of the degraded area in the respective types of the communes (ha).

| Percentiles | Cities with County Rights | Urban Communes | Urban–Rural Communes | Rural Communes |
|---|---|---|---|---|
| 1. | 113.00 | 29.00 | 40.00 | 108.00 |
| 2. | 333.60 | 68.60 | 96.40 | 252.00 |
| 3. | 514.80 | 120.00 | 193.00 | 643.50 |
| 4. | 631.00 | 165.20 | 351.80 | 1110.00 |
| 5./median | 1030.00 | 210.00 | 890.00 | 1762.50 |
| 6. | 1395.60 | 281.00 | 1831.20 | 2543.00 |
| 7. | 1708.30 | 395.60 | 3142.60 | 3533.50 |
| 8. | 2697.60 | 556.80 | 4845.60 | 4947.00 |
| 9. | 3853.20 | 936.00 | 7966.60 | 7201.00 |
| 10. | 11,209.00 | 5130.00 | 30,000.00 | 23,488.00 |
| Average: | 1691.98 | 405.11 | 2884.44 | 2970.71 |

The data in the Table above show that the largest degraded areas are noted in the cities with county rights and rural communes. On the other hand, the greatest concentration of small, degraded areas can be observed in the urban communes, of which in 90% of cases the degraded areas do not exceed 1000 ha. In other types of the local government units, areas were designated in 50% of the cities with county rights, slightly more than in 40% of the urban–rural communes and in 30% of the rural ones. Additionally, in order to compare the scale of this differentiation, the relation of the average value and the value for the first percentile in a group and the relation of the highest value in a group to the value for the first percentile were calculated (Table 7).

**Table 7.** Differentiation of the surface of degraded areas in the respective types of the communes.

| Description | Cities with County Rights | Urban Communes | Urban–Rural Communes | Rural Communes |
|---|---|---|---|---|
| Average/lowest percentile | 14.97 | 13.97 | 72.11 | 27.51 |
| Highest percentile/lowest percentile | 99.19 | 176.90 | 750.00 | 217.48 |

The greatest disproportions were observed in the urban–rural communes. In the case of the cities with county rights and urban communes, there is a similar relation between the average value and the value for the first percentile for these local government units. The discrepancy is visible for local government units with larger degraded areas. In the group of cities with county rights, the value for the last percentile is approximately one hundred times the size of the degraded areas for the first percentile. In the case of the urban communes, it is almost 180 times bigger.

The delimitation of a degraded area is followed by a decision on the size of the area that will become a revitalisation area. According to Art. 10 para. 1 of the Revitalisation Act, such area should be characterised by a particular concentration of negative phenomena and importance for local development. In order to limit implementing dispersed activities on a large area, the authorities introduced in Art. 10 (2) restrictions on the size of the area and its population. The revitalisation area cannot exceed 20% of the commune's area and 30% of its inhabitants. The largest number of revitalisation areas (RA) were established in rural communes: 59.6% of the total surface of the RAs. Rural communes also dominate in terms of the average RA surface, although only slightly smaller, on average, revitalisation areas were designated in cities with county rights. The total surface of revitalisation areas in the respective types of the communes is presented in the Table below (Table 8):

**Table 8.** Surface of the revitalisation areas by the type of the commune.

| Type of the Commune | Total Surface of the Revitalisation Areas (ha) | Share of the Revitalisation Areas in the Area of a Commune (%) |
| --- | --- | --- |
| Urban commune | 88,710 | 4.10% |
| including cities with county rights | 43,322 | 5.80% |
| Urban–rural commune | 379,826 | 3.62% |
| Rural commune | 689,982 | 3.56% |
| Total: | 1,158,518 | 3.62% |

With regard to rural areas, the extensive variant of delimiting areas results from a significant dispersion of buildings. In the case of the cities with county rights, large revitalisation areas result in the dispersion of activities over a larger area of intensive development, and, therefore, probably a low support per resident [88].

For the purpose of the article, data on the size of the revitalisation areas in all types of the communes according to percentiles were analysed (Table 9).

**Table 9.** Percentiles and the average size of the revitalisation area in the respective types of the communes (ha).

| Percentiles | Cities with County Rights | Urban Communes | Urban–Rural Communes | Rural Communes |
| --- | --- | --- | --- | --- |
| 1. | 178.30 | 30.80 | 40.40 | 49.00 |
| 2. | 271.20 | 64.60 | 73.00 | 131.00 |
| 3. | 360.80 | 96.20 | 99.60 | 206.00 |
| 4. | 442.20 | 121.80 | 141.00 | 333.00 |
| 5./median | 544.00 | 148.00 | 202.00 | 643.50 |
| 6. | 631.00 | 172.00 | 297.00 | 990.00 |
| 7. | 743.00 | 213.00 | 449.40 | 1344.00 |
| 8. | 875.00 | 282.40 | 1180.80 | 1758.00 |
| 9. | 1522.40 | 396.60 | 2006.00 | 2292.50 |
| 10. | 2387.00 | 2898.00 | 30,000.00 | 9494.00 |
| Average: | 676.91 | 207.25 | 2884.44 | 990.69 |

The above Table shows that the largest revitalisation areas are determined in the cities with county rights. On the other hand, in the rural communes, the greatest increase in the surface occurs between the subsequent percentiles. As in the case of the degraded areas, the greatest concentration of small revitalisation areas can be observed in the urban communes. In 90% of them revitalisation areas do not exceed 400 ha. In the case of other types of local government units, areas with such size were designated in 30% of the cities with county rights, in 60% of the urban–rural communes and in 40% of the rural communes. Additionally, to compare the scale of the differentiation, the relation of the average value for a group and the value for the first percentile in a group and the relation of the highest value in a group to the value for the first percentile were calculated (Table 10).

**Table 10.** Differentiation of the surface of the revitalisation area in the respective types of the communes.

| Description | Cities with County Rights | Urban Communes | Urban–Rural Communes | Rural Communes |
| --- | --- | --- | --- | --- |
| Average/lowest percentile | 3.80 | 6.73 | 71.40 | 20.22 |
| Highest percentile/lowest percentile | 13.39 | 94.09 | 742.57 | 193.76 |

The greatest disproportions, as in the case of the degraded areas, were observed in the urban-rural communes. The relation of the cities with county rights and urban communes is different than in the case of the degraded areas. In that case, there was a similar relationship between the average and the value for the first percentile, and significant discrepancies in differentiation were visible for local government units with larger degraded areas. In

the case of the revitalisation areas, the ratio of the average value to the value for the first percentile was twice as high in the case of the urban communes. This proves the high discrepancy; however, the revitalisation areas are still much smaller in the urban communes than in the cities with county right. The revitalisation areas in the cities with county rights are not as differentiated as in the case of the degraded areas. The value for the last percentile of the communes in this group was approximately one hundred times the size of the degraded areas for the first percentile, but total for revitalisation areas only 14 times. In the case of the urban communes, the values are still higher. The revitalisation area in the last percentile is almost 100 times bigger than in the first percentile. It should be emphasised, however, that the last percentile in the case of the urban communes is unreliable due to a large peak in value compared to the previous nine.

In the next step, the number of the residents inhabited the revitalisation areas in different types of the communes according to percentiles was analysed (Table 11).

**Table 11.** Percentiles and the average population of the revitalisation area in the respective types of the communes in (pers).

| Percentiles | Cities with County Rights | Urban Communes | Urban–Rural Communes | Rural Communes |
|---|---|---|---|---|
| 1. | 5051 | 1178 | 953 | 454 |
| 2. | 10,935 | 1801 | 1397 | 733 |
| 3. | 14,051 | 2941 | 1802 | 949 |
| 4. | 18,931 | 4006 | 2151 | 1110 |
| 5./median | 26,997 | 4918 | 2559 | 1270 |
| 6. | 33,201 | 5747 | 3090 | 1474 |
| 7. | 35,042 | 7152 | 3771 | 1785 |
| 8. | 47,787 | 9492 | 4668 | 2171 |
| 9. | 56,820 | 11,829 | 6136 | 2800 |
| 10. | 152,292 | 30,576 | 14,340 | 11,637 |
| Average: | 32,392 | 6006 | 3182 | 1521 |

The Table above shows that the most populous revitalisation areas are in the cities with county rights. In this group, the largest increase in the number of people between consecutive percentiles is also noticed. The revitalisation areas were three times more densely populated than degraded areas, so the revitalisation intervention focuses on a much smaller area but including as many of inhabitants as possible. The average share of the revitalisation area in a total area of a commune varies between 3.2 and 5.8% depending on a type of the commune, while the share of the number of the inhabitants of the revitalisation area in the entire population of the commune oscillates around 20% [88]. The step towards focusing on a limited area does not therefore go hand in hand with the concentration of support in relation to the needs of its recipients. The implemented projects concentrate on the improvement of the quality of public buildings and centrally located public spaces, and thus serve all city residents, not solving the problems of the revitalisation area. This is one of important arguments confirming the thesis put forward in the introduction to the article that the goal of local authorities is an extensive land use policy.

To compare the scale of differentiation of the number of the revitalisation area inhabitants, the relashionship of the average for a group and the value for the first percentile in a group was calculated, as well as the relationship of the highest value in a group to the value for the first percentile (Table 12).

The largest disproportions were observed in the cities with county rights; however, the scale of differentiation is much smaller than in the case of the degraded areas or revitalisation areas. The urban–rural and rural communes are similar in terms of the relationship between the highest and the lowest number of the revitalisation area inhabitants, and the range between the largest and least numerous revitalisation areas is similar in the urban and rural communes. It is worth emphasising that both in the urban–rural and rural communes, the revitalisation areas were limited in the inhabitants' number to a greater extent.

**Table 12.** Differentiation of the number of the revitalisation area inhabitants in the respective types of the communes.

| Description | Cities with County Rights | Urban Communes | Urban–Rural Communes | Rural Communes |
|---|---|---|---|---|
| Average/lowest percentile | 6.41 | 5.10 | 3.34 | 3.35 |
| Highest percentile/lowest percentile | 30.15 | 25.95 | 15.05 | 25.63 |

*3.3. The New Investment in a Sewage Network and the Large-Scale Degraded and Revitalisation Areas—Contradiction and Coexistance*

An extensive spatial management and land use policy model can be observed in Poland. That means that local governments create conditions favouring area-consuming land use approach [28]. Such unconcern about the scope of new housing development is accompanied by another phenomenon. Local authorities, choosing together with inhabitants, the range of the degraded and revitalisation areas, try to enable the largest possible areas and the largest population to have chance to obtain external subsidies for revitalisation (from regional or national grants). This results in the differenciation of the effects of revitalisation activities.

This part of the article focuses on the coexistence of two contradictory tendencies in the Polish local governments' land use policy, i.e., the expansion of sewage networks along with urban sprawl, and the designation of extensive degraded and revitalisation areas. The analysis was carried out on a regional basis, taking into account three types of communes: urban, urban–rural, and rural. The analysis did not include the cities with county rights due to their dominance over other types of the communes in terms of much larger investments in the development of sewage networks, as well as significantly larger degraded and revitalisation areas in relation to other types of the communes.

Calculations of the correlation coefficient for the increases in the length of sewage networks and the size of degraded and revitalisation areas for the rural and urban–rural communes led to the rejection of the hypothesis about the relationship for these types of the communes (Table 13). In the case of the urban communes, however, apart from two Voivodeships (Opolskie and Świętokrzyskie, where the relationship turned out to be statistically insignificant), the results showed a significant relationship.

**Table 13.** Values of the correlation coefficient for the increase in the length of sewage networks and the size of the degraded and revitalisation areas.

| Voivodeship | Correlation Coefficient X | |
|---|---|---|
| | $Y_1$ [1] | $Y_2$ [2] |
| Dolnośląskie | 0.89 | 0.46 |
| Kujawsko-Pomorskie | 0.88 | 0.95 |
| Lubelskie | 0.51 | 0.52 |
| Lubuskie | 0.70 | 0.98 |
| Łódzkie | 0.72 | 0.93 |
| Małopolskie | 0.94 | 0.82 |
| Mazowieckie | 0.34 | 0.78 |
| Podlaskie | 0.89 | 0.88 |
| Pomorskie | 0.56 | 0.79 |
| Śląskie | 0.42 | 0.37 |
| Warmińsko-Mazurskie | 0.42 | 0.76 |
| Wielkopolskie | 0.69 | 0.44 |
| Zachodniopomorskie | 0.96 | 0.52 |

[1], [2] $\alpha = 0.05$, *p*-value $\leq 0.01$.

On the basis of the presented results, it can be concluded that in the urban communes where the expansion of sewage networks can be noticed there are also large degraded

and revitalisation areas. This is an important information, taking into account the lowest differentiation in this type of the communes in relation to the other types.

On the other hand, the values of the correlation coefficient for the increase in the length of sewage networks and the number of inhabitants of the revitalisation areas show a relationship regardless of the type of the commune (Table 14).

**Table 14.** Values of the correlation coefficient for the increase in the length of sewage networks and the number of inhabitants of the revitalised areas by the Voivodeship and type of the commune.

| Voivodeship | Correlation Coefficient $X/Y_3$ | | |
|---|---|---|---|
| | Urban Communes [1] | Urban–Rural Communes [2] | Rural Communes [3] |
| Dolnośląskie | 0.86 | 0.45 | 0.41 |
| Kujawsko-Pomorskie | 0.93 | 0.72 | 0.53 |
| Lubelskie | 0.82 | 0.02 | 0.44 |
| Lubuskie | 0.98 | 0.68 | 0.16 |
| Łódzkie | 0.96 | 0.50 | 0.77 |
| Małopolskie | 0.94 | 0.19 | 0.20 |
| Mazowieckie | 0.98 | 0.45 | 0.50 |
| Podlaskie | 0.96 | 0.65 | 0.88 |
| Pomorskie | 0.78 | 0.34 | — [4] |
| Śląskie | 0.56 | 0.86 | 0.31 |
| Warmińsko-Mazurskie | 0.81 | 0.16 | — [5] |
| Wielkopolskie | 0.93 | 0.60 | 0.36 |
| Zachodniopomorskie | 0.91 | 0.44 | 0.37 |

[1] $\alpha = 0.05$, p-value $\leq 0.01$; [2], [3] $\alpha = 0,05$, p-value $\leq 0.001$; [4] There are no revitalisation areas in the rural communes in the Pomorskie Voivodeship.; [5] A low number of rural communes with the revitalisation areas makes the analysis pointless.

As in the case of the above-discussed relationship, the strongest correlation is observed for urban communes. On the other hand, for urban–rural and rural ones, the strength of correlation is a reflection of regional guidelines for the delimitation of revitalisation areas besides the Lubelskie Voivodeship where in the urban–rural communes the stagnation in access to the sewage network is deepening, which is not related to the tendency of urban sprawl. A separate study should be devoted to a detailed explanation of the reasons for this differentiation.

*3.4. Heating Network Routes and the Directions of Housing Development—The Example of the Dolnośląskie Voivodeship*

In the face of the current challenges related to the increasing pressure on environmental resources, including space which should be treated as a non-renewable or hard-to-renew resource, special attention should be paid to the directions of spatial development of cities as well as the energy conservation principle. In addition to the problems related to climate change and the need to create settlements resistant to these changes, there is a problem of air pollution and low emissions, such as PM 2.5 and PM 10 dust pollutants, and polycyclic aromatic hydrocarbons affecting the health of residents, as well as nitrogen oxides mostly due to means of transport. Most of the air pollutants come from the housing and service areas which are responsible for the emission of 66% of PM10 suspended dust, 84% of PM2.5 and 98.4% of the carcinogenic benzo[a]pyren B(a)P [89].

Therefore, taking into account the fact that cities are equipped with district heating networks, the air pollution concentration can be reduced by connecting residential and service buildings to the existing central heating networks, which is followed by the elimination of dispersed solid fuel installations [90–92]. The outdated coal-fired installations which, in practice, are used to burn also other combustible solid materials, including household waste, become the main polluter. Therefore, a rational solution would be to concentrate housing development in the city cores and use the existing heating system potential [92]. That could happen if the cost of air pollution and of the impact of harmful substances on the health of residents were internalised and taken into account when balancing the costs of the

city development and granting permits for the investment implementation. Unfortunately, that is not the case, which results in the chaotic and extensive use of urban space as well as the suburban area for the development of housing and service areas (HSAs). The Table below presents the list of the communes with the increased length of the networks and of the connections to buildings in the Dolnośląskie Voivodeship. These communes are ranked starting with the highest increase in the length of the network in 2017–2019 (Table 15). The analysis is limited to the abovementioned period due to the availability of data from Statistics Poland.

**Table 15.** Increase in the length of the heating network in the communes of the Dolnośląskie Voivodeship in 2017–2019 [km].

| Name | Increase in the Length of the Heating Network | Increase in the Length of the Connections to Buildings |
|---|---|---|
| Wrocław (1 [1]) | 9.8 | 13.8 |
| Świdnica (1) | 7.1 | 0.0 |
| Jelenia Góra (1) | 2.6 | 1.4 |
| Dzierżoniów (1) | 1.9 | 0.9 |
| Bolesławiec (1) | 1.9 | 0.2 |
| Złotoryja (1) | 1.7 | 0.2 |
| Głogów (1) | 1.5 | 1.6 |
| Oleśnica (1) | 1.3 | 0.8 |
| Zawidów (1) | 0.7 | 0.1 |
| Lubań (1) | 0.7 | 0.4 |
| Stronie Śląskie (3) | 0.4 | 0.0 |
| Bielawa (1) | 0.4 | 0.4 |
| Legnica (1) | 0.4 | −2.3 |
| Świebodzice (1) | 0.3 | 0.4 |
| Oława (1) | 0.3 | 1.6 |
| Wałbrzych od 2013 (1) | 0.3 | 0.8 |
| Jelcz-Laskowice (3 [2]) | 0.2 | 0.2 |
| Lubin (1) | 0.1 | 2.4 |
| Ząbkowice Śląskie (3) | 0.1 | 0.4 |
| Zgorzelec (1) | 0.1 | 0.1 |

[1] Urban commune; [2] Urban–rural commune.

It is worth noting that in Świdnica, with the second largest heating network, this increase did not result in new connections to buildings. At the same time, Świdnica is one of the communes with the longest sewage network in the region, which has undergone significant expansion in recent years (over 30 km in the last ten years).

Based on the results of the ITD research, several cities from the Dolnośląskie Voivodeship were selected for a cross-sectional analysis.

The following examples from the cities from the Dolnośląskie Voivodeship illustrate this phenomenon as well as the continuing negative trend confirmed by a low increase in the connections to the heating networks in 2017–2019 (see Table 16 above).

In the largest city in the region, Wrocław (Figure 1), inhabited by about 650,000 inhabitants, the district heating network covers mainly the central part of the city, i.e., the city centre, together with areas subject to the necessary revitalisation intervention due to the noticeable degradation of the housing buildings as well as the partial pauperisation of the society. However, in the Study of the conditions and directions of spatial development the areas intended for housing, service, or mixed-function with a predominant share of the residential or residential and service function (hereinafter referred to as housing and service areas, HSAs) have been delimited very extensively, enabling the development of areas remote from the city core. Only 41.8% of residential areas are within the range of the heating network. The heat supply indicator of residential buildings is also low and amounts to only 27.3% of the buildings within the district heating network. Intensive investments in areas remote from the city centre will imply in the future additional shuttle

migrations in the city and its suburbs, which will contribute to the energy consumption of the urban tissue.

**Table 16.** Access to the district heating networks in the selected cities of the Dolnośląskie Voivodeship in view of their land use policy.

| HAS Surface (ha) | HSA Surface within the Range of the Network (ha) | Number of Buildings in the Range of the Network (psc.) | Number of Buildings Connected to the Network (psc.) | Percentage of Buildings Connected to the Network (%) | HSA Share in the Range of the Network (%) | Name of the City |
|---|---|---|---|---|---|---|
| 13,163 | 5502 | 28,041 | 7644 | 27.26 | 41.80 | Wrocław |
| 2553 | 843 | 5251 | 657 | 12.51 | 33.01 | Jelenia Góra |
| 2100 | 279 | 1534 | 675 | 44.00 | 13.27 | Wałbrzych |
| 1905 | 718 | 4094 | 1540 | 37.62 | 37.70 | Lubin |
| 1770 | 1209 | 7728 | 1071 | 13.86 | 68.30 | Legnica |
| 1039 | 512 | 3456 | 508 | 14.70 | 49.22 | Świdnica |
| 964 | 651 | 3646 | 773 | 21.20 | 67.55 | Głogów |
| 860 | 31 | 255 | 33 | 12.94 | 3.65 | Nowa Ruda |

HSA—housing and service area.

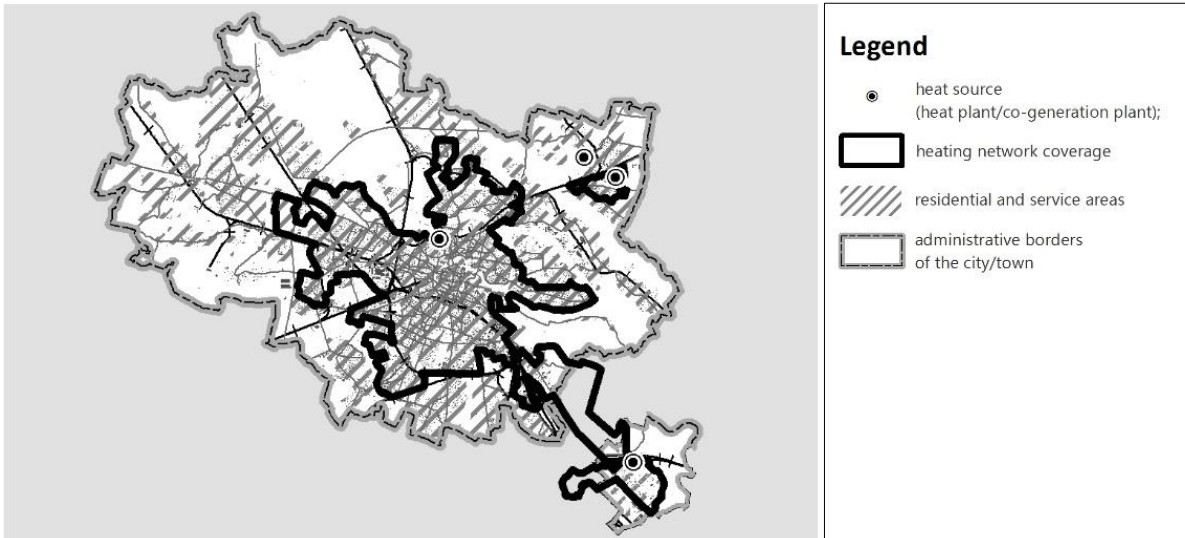

**Figure 1.** Areas intended for housing and service development, and the range of the district heating network in Wrocław and the neighbouring town of Siechnice. * HSAs—housing and service areas.

In Wałbrzych (approx. 110,000 inhabitants), the second largest city of the Voivodeship, the heating network covers only two housing estates located in the northern part of the city: Piaskowa Góra and Podzamcze quarters (Figure 2.). The housing estates were built in the large panel technology in the second half of the 20th century. However, 56% of neighbouring buildings, situated outside the housing estates, are not connected to the district heating network. The entire city is located in the mountain valleys, which certainly make the development of the heating, sewage, and water supply networks difficult. Only 13.3% of the housing and service areas indicated in the planning documents are within the range of the heating network. Thus, search for directions of the city's spatial development that would ensure heating systems based on non-emission or low-emission technologies is still required.

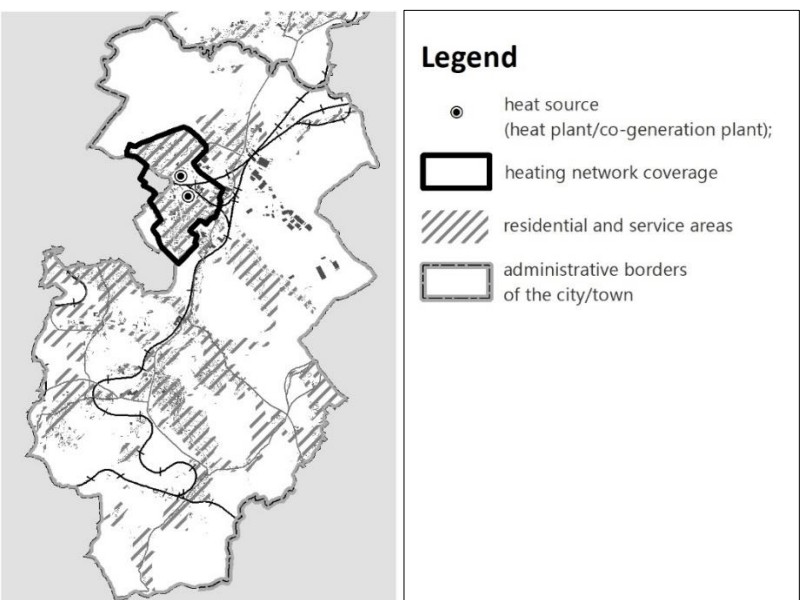

**Figure 2.** Areas intended for housing and service development, and the range of the district heating network in Wałbrzych. * HSAs—housing and service areas.

In Legnica, the city with almost 100,000 inhabitants, housing and service areas are concentrated in the central and eastern part of the city, forming a compact space for the building development, well correlated with the range of the heating network (Figure 3.). The district heating network covers 68% of the housing and service areas indicated in the city planning documents. However, this potential is not used. Only 13.9% of the buildings within the range of the heating network are connected to it.

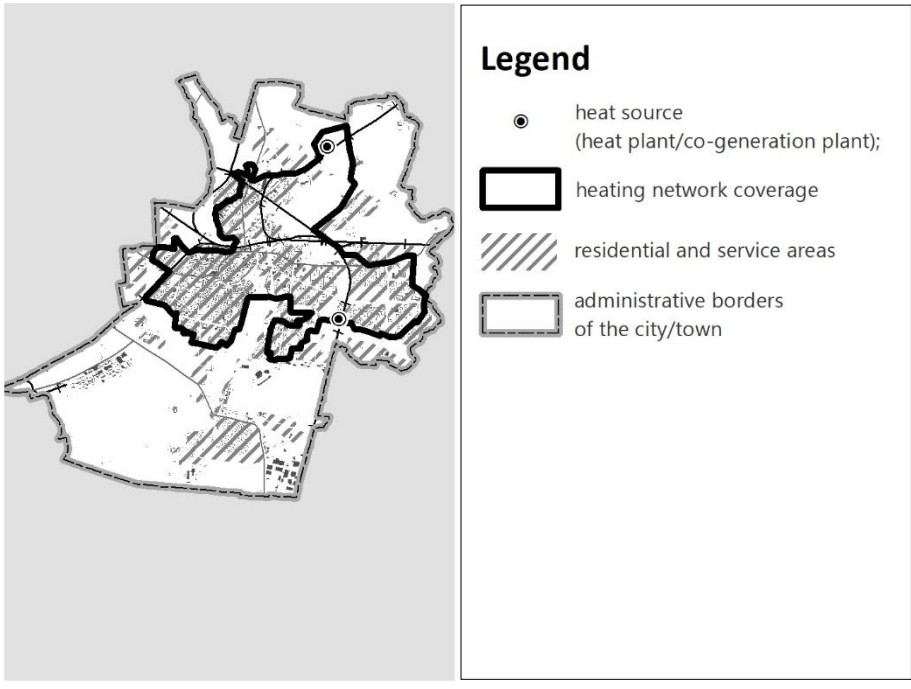

**Figure 3.** Areas intended for housing and service development, and the range of the district heating network in Legnica. * HSAs—housing and service areas.

The spatial development policy of Jelenia Góra, the city of almost 80,000 inhabitants, also features large housing and service areas beyond the reach of the heating network. However, due to geomorphological conditions and the city's location in a mountain valley, urban development is naturally limited by an orographic barrier (Figure 4). The heating network covers only 33% of the housing and service areas, including existing and planned developments foreseen in the city plans. Within the area of the district heating network range, only 12.5% of the buildings are connected to the network. In the case of Jelenia Góra and Wałbrzych, the location in the mountain area and the general use of individual heat sources for solid fuels are the cause of low air quality in winter. Elimination of pollution sources can take place through the use of new technologies, but also by using the potential of the existing heating network. Moreover, this conclusion applies to all the cases discussed in the article.

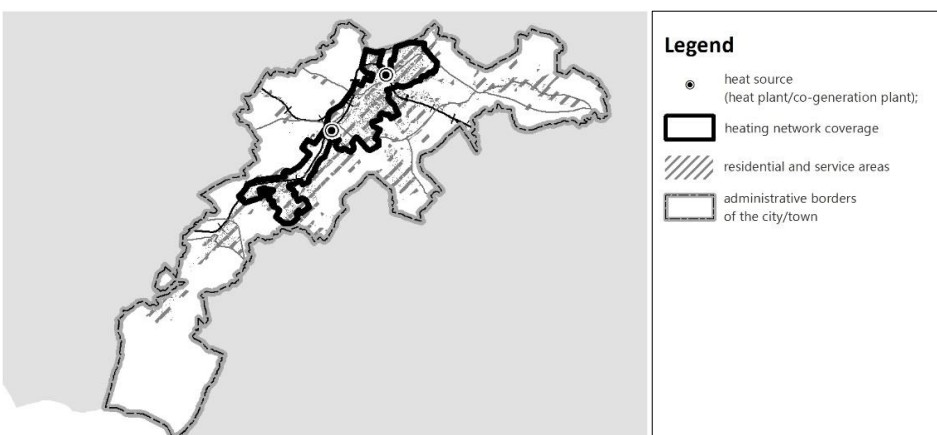

**Figure 4.** Areas intended for housing and service development, and the range of the district heating network in Jelenia Góra. * HSAs—housing and service areas.

Głogów inhabited by almost 67,000 inhabitants, similarly to Legnica, has a very high ratio of 67.5% of the housing and service areas within the range of the heating network (Figure 5). The high concentration of inner-city developments is the result of the reconstruction of the urban fabric after the WWII, while maintaining the historic urban structure. However, in the city only 21% of the buildings are located within the range of the district heating network. The vast area in the western part of the city is excluded from development as it is occupied by a copper smelter. It is also one of the reasons of placing residential areas in the eastern part of the city and, partially, in the suburban area.

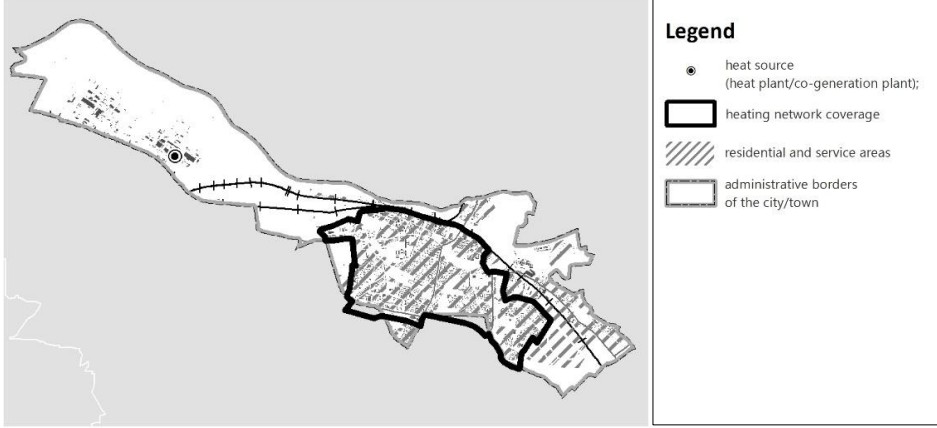

**Figure 5.** Areas intended for housing and service development, and the range of the district heating network in Głogów. * HSAs—housing and service areas.

In Świdnica, inhabited by almost 57,000 inhabitants, housing and service areas have been delimited concentrically around the old town (Figure 6.). The district heating network has been developed in the northern part and covers almost half of these areas. The heating network is mainly connected to new multi-family housing estates, including those built since the 1960s. Despite the extensive range of the network, less than 15% of the buildings use its potential, which has not changed in recent years. It is worth noting that, unlike the neighbouring Wałbrzych, Świdnica, this middle-size city develops already its distinct residential suburbia zone, beyond the administrative borders. This process strengthens the extensive development of the city.

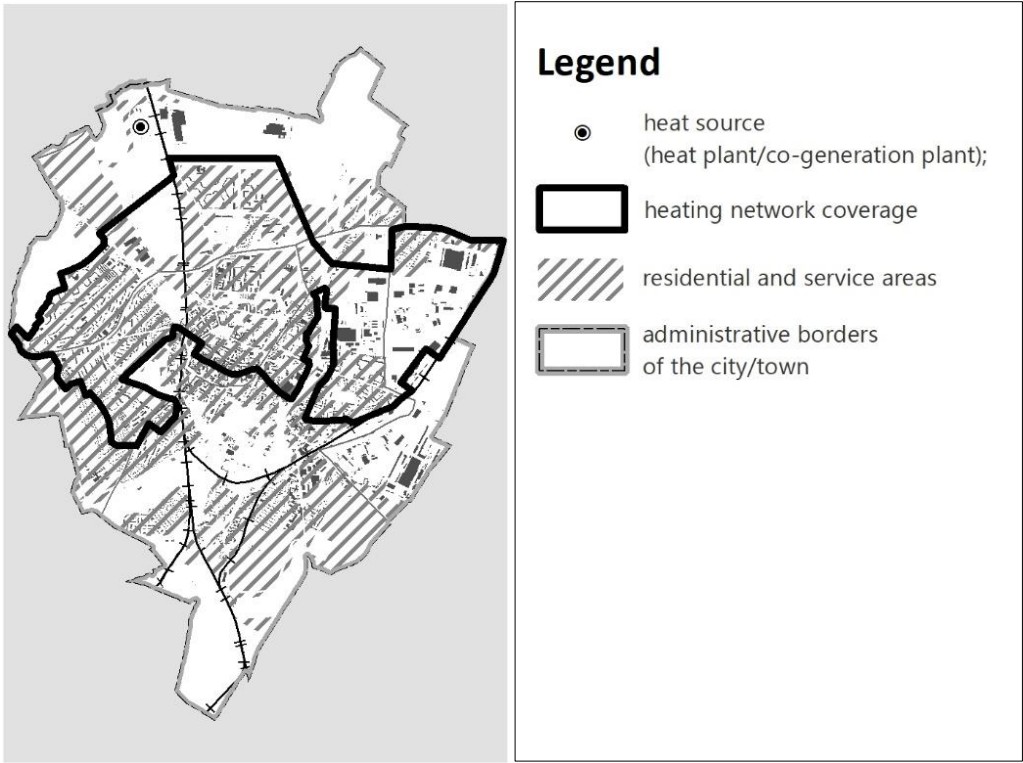

**Figure 6.** Areas intended for housing and service development, and the range of the district heating network in Świdnica. * HSAs—housing and service areas.

Lubin is a young city, whose development is mainly related to the important mining and metallurgical works. The copper industry, in the city with over 70,000 inhabitants, resulted in a dynamic development of residential areas, and in the increase in the wealth of the inhabitants, and, as a consequence, led to suburbanisation processes. While in the larger cities of the region (Wrocław, Legnica, Jelenia Góra), this process is part of the well-known and replicated Western model of the suburban development, in the case of Świdnica and Lubin, the communes with a slower pace of life and economy, the process of urban sprawl is a new phenomenon, generated by the increase in the wealth of the society. In the Lubin's spatial development policy, extensive, for the city's conditions, housing and service areas have been projected, 38% of which are connected to the heating network (Figure 7).

Nowa Ruda is the last of the examples of use of the potential of heating networks in the cities of the Dolnośląskie Voivodeship. The city has been struggling with air quality problems for years, as in the autumn and winter seasons, the limits of air pollutants are exceeded. That is caused by surface emission from the housing and service areas. The spatial policy of the city with less than 22,000 inhabitants, due to the small range of the heating network, has located over 96% of the housing and service areas outside the range

of the network. At the same time, only 13% of buildings within its range use its potential (Figure 8).

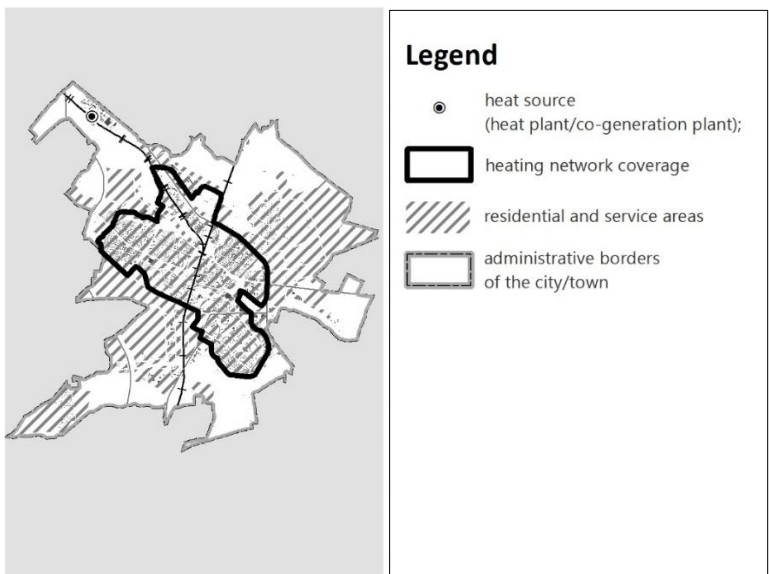

**Figure 7.** Areas intended for housing and service development, and the range of the district heating network in Lubin. * HSAs—housing and service areas.

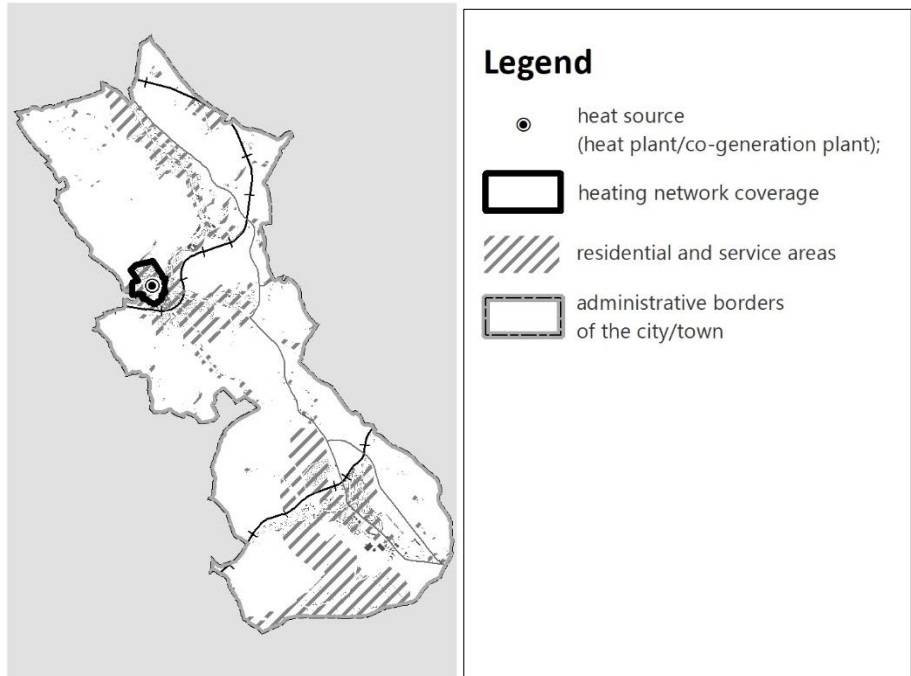

**Figure 8.** Areas intended for housing and service development, and the range of the district heating network in Nowa Ruda. * HSAs—housing and service areas.

## 4. Discussion

The coexistence of development sprawl and dispersal of revitalisation activities on large areas was confirmed in the previous section by the statistical dependence for urban communes. At the same time, the tendencies to designate large degraded and revitalisation areas in the urban–rural and rural communes were discussed. Rural areas dominate in the structure of the revitalisation areas due to the difficulty of delimiting a degraded area

and revitalisation area in units smaller than sołectwo (the smallest administrative unit in Poland) or individual villages. Such areas are generally qualified for revitalisation, despite the fact that the rural communes cannot effectively conduct revitalisation, because this process and its inventory are not adapted to their specificity, but for dense developed downtown areas [88]. In some of these communes, there is also a correlation between the surface of the degraded and revitalisation areas and the dispresed expansion of the sewage network. This mainly applies to the urban–rural and rural communes in the functional areas of the Voivodeship capital cities and, often, also to other cities with county rights. A separate category, with a similar relationship, are rural communes adjacent to a city. For detailed case study analyses it would be necessary to conduct separate GIS study to select communes where the dispersed expansion of the sewage network coexists with the scattered residential areas.

The presented results contribute to the research on the symptoms of spatial chaos. Although the article concern Poland, the outcomes should be considered in a broader context of previous studies on urban sprawl in other countries or comparative studies and as a part of the discussion on policies, governance and management models oriented at encouraging land-consumption behaviours that have been going on in the urban literature and practice for several decades [9,10,14]. Restrictive and adaptive land use policy approaches are still on struggle [28,31]. The restrictive approach supports the development of planning documents and land use policy under the principles of sustainable development, emphasises the negative impacts of urban sprawl on environment and public health. Its result is a big scope of literature on the costs of urban sprawl [93–97], urban containment policies limiting new residential development, e.g., city boundaries or growth management plans [49,98,99]. The adaptive approach, on the contrary to the restrictive one, advocates a laissez-faire attitude of local authorities, undermining the validity of the negative effects of sprawl [100–105]. Some literature suggests even that compactness may also have negative impacts on social and ecological dimensions, particularly open space preservation and biodiversity, traffic flow, health, and well-being. The evidence comes from the analysis of job accessibility and pollution reduction. It suggests that policy interventions promoting compactness should deliver healthier and greener cities [106]. The criticism against compact cities refers to the problems of the Global South where compactness of the cities has rather negative impact on access to jobs, services, and amenities than in the global North [105]. The discrepancy of experiences and the characteristics of the challenges of the Global South and North indicate that the countries of the Global North should limit the spread of development. Despite evidence of the negative impact of urban sprawl, the process cannot be effectively mitigated. There is even so called suburban paradox, pointing suburban livability versus huge costs of urban sprawl and the ever-green desire of big group of inhabitants to live in the cities' outskirts all over the world [107]. This article contributes to the source literature devoted to identifying the symptoms of urban sprawl and presenting traps of the extensive models of land use policy.

This type of research is especially needed as a basis for educational activities for local government authorities in Poland because of the progressing spatial chaos. Other countries have also struggled with similar trends. It is worth noting the recent retreat of Australian cities from the earlier North American model of suburban development featuring low-population, car-dependency, and greenfield development. The negative social, economic, and environmental consequences related to the strengthening of such model became visible already in the 1990s, followed by the policy of compact cities, initially mainly in the slogans of expert circles. Gradually, planners also began to incorporate them into long-term metropolitan planning strategies. This transition from "suburban" to "urban" style is still a challenge, but it has been featured in metropolitan planning strategies since the 2000s, even in large cities such as Melbourne and Sydney. The target city population of 70% for new housing developments in these cities is a big challenge for the main stakeholders involved in Australian urban development are: state and local governments, the real estate industry, and aging suburban residents. Strategies for the

renovation and revitalisation of central districts along with stopping new investments in the suburbs [108] are of key importance for achieving such transformation.

One of the significant costs of urban sprawl is excessive expenditure on the new infrastructure in the leapfrog development [49,99]. The results of the multiple research concern all types of infrastructure [109–112]; however, for the purposes of this article, the cost of the sewage network has been selected. It was not a random choice. The problem of water consumption in the scattered housing structure is the subject of extensive literature studies [113–115]. In many European cities, compact cities change to more dispersed one [116] what has a negative impact on water (both clean and waste) management. From this perspective, Polish statistical data showing the expansion of the water supply and sewage network do not differ from the pan-European trends. However, two other issues are worth noting. The first is the ability to internalise the costs of providing and maintaining sewage infrastructure in the conditions of urban sprawl. In the Spanish literature for instance, particular attention is paid to the possibility of recovering increased expenditure on water supply in its price. Moreover, the costs of providing new infrastructure are internalised with the Municipal Urbanisation Tax, a tax levied on newly built buildings. The incomes from this particular tax are used to finance the construction, maintenance, and modernisation of municipal infrastructure [117]. Such a solution is lacking in Poland, taking into account the ineffectiveness of the Polish adiacency fee [54]. The second problem is the counter-effectiveness of expenditure on a new sewage system, strongly emphasised in the British literature. The costs of modernising the existing one exceed the budget of the cities, and the new one should not be undertaken before modernisation ends [114]. In Polish conditions, due to many years of underinvestment [82], the abandon of the construction of a new sewage network cannot be taken seriously into consideration. New investments should be planned taking into account the efficiency criterion, i.e., first for densely populated districts, and only then in scattered housing areas.

Another topic discussed in this article is the stimulation of the tendencies to urban sprawl through the degradation of the city centres. In literature the term "holding capacity" refers to city centres, whose functions, history, and the spirit of a place are sufficient to retain or attract residents [118–120]. The outflow of downtown residents caused by poor housing conditions and blight has been the subject of analysis in the context of suburbanisation for many years, especially "flight from blight" concept [48,50]. These issues are still rarely co-analysed in the Polish literature. Meanwhile, revitalisation, which in Polish conditions is progressing dynamically throughout the country [88], may be one of the remedies for sprawl. Maybe, but not necessarily, because as the research presented in the article shows, the revitalisation areas are too large to be subject to change in a short time. Revitalisation activities are too fragmented. Therefore, when in two contradictory and simultaneous processes, as shown in the article, funds are spent on the development of dispersed infrastructure and the concentration of revitalisation activities, suburbanisation tendencies are strengthened.

In Poland, it is still rare that communes lead consciously the land use policy limiting sprawl by coordinating spatial planning (limiting land for housing development and consciously carrying out revitalisation activities). Wałbrzych is one of the exceptions. In connection with the pilot project in the field of revitalisation in Włabrzych, planning documents were developed in order to integrate spatial policy with the revitalisation process. Due to the preparation of a new Study of the conditions and directions of spatial development, the oversupply of housing development areas for the entire city was restricted, in particular in peripheral areas, based on the analysis of built-up areas in relation to the current demographic forecasts. Moreover, arrangements to support shaping of public spaces for respective districts were made and heritage policy to build up the city's identification was defined. This is but an avantgarde, not a standard among Polish cities.

The Wałbrzych's different approach probably results from the challenges that the city has had to deal with in recent years. Wałbrzych has underwent difficult years, both in the 1990s due to the liquidation of mines and the textile industry, and in 2000s, when

in 2003 for eleven years it lost the position of a city with county rights (it was not any more a unit combining the rights and obligations of a commune and a poviat being a higher-level local government unit). The latter also resulted in a decrease in the city's income. In 2003, compared to 2002, total income decreased by PLN 70 million, i.e., by a quarter. The dynamic increase in income after regaining the former position in 2013 has shown how much the city lost during that stagnation period. The average income in the years 2003–2012 amounted to PLN 281.5 million, while after 2013 it has come to PLN 559 million. The commune could not take full advantage for financing revitalisation from the previous Regional Operational Programme, because most of the property that required urgent investments did not belonged to the commune but to the poviat. As a consequence, some projects were postponed to 2008–2015, and even to 2016–2019. Therefore, despite a relatively large, degraded area, revitalisation activities are concentrated on a smaller area, where most of the planned projects are implemented, including projects by private entities [121].

The heating system in the cities selected for the analysis is not optimally used. The spatial development policy of housing and service areas, defined in the Studies of conditions and directions of spatial development, in most of the analysed areas, rarely takes into account the possibilities of using heating systems to ensure heat supply. Thus, the energy efficiency of the planned urban structure is low and based on individual choices, which increases the consumption of fossil fuels and reduces the efficiency of the urban system as a whole. In the context of emerging new technologies in heating and the need to diversify energy sources, as well as in the context of decarbonisation of the economy, including energy systems, the functioning of the heating networks, powered by fossil fuels, is becoming a civilisation problem in cities.

However, the energy transformation in cities should take place in an evolutionary manner, and space management should be based on the use of existing resources, so of the potential of the existing heating networks. Neglecting the heating network issues in the concepts and adopted planning documents results in a poor air quality, confirmed by annual assessments. Such problem occurs in urban and suburban areas with individual heating systems. The use of a heating network could partially solve the environmental problems, which is why the solutions regarding the spatial development of cities should always take into account a local context.

Another illustration of extensive land use policy in Poland is the lack of coordination of spatial development and the use of municipal heating systems in the Dolnośląskie Voivodeship in Poland. The relationship of spatial planning and development, as well as the use of the existing energy infrastructure is emphasised by studies comparing the policies of the cities of Vancouver, Hong Kong, Oslo, and Oakland [122]. Asaporta and Nadin raise arguments for taking into account the energy transformation in urban planning and their urban form, which has and will have a strong impact on the medium and long-term future of the energy transformation and vice versa. Further, Danish researchers point out to the need to plan urban development taking into account the energy strategy [123]. They indicate that communes (including cities) should have the knowledge and necessary tools to conduct energy planning. This argues for the necessity to coordinate development policies and greater awareness of the impact of small, local-scale political decisions on the achievement of global goals, including increasing the energy efficiency of cities.

There may be a concern that adapting the spatial structure to the network infrastructure developed in the past on the basis of heating technology associated with the combustion of fossil fuels is not the right direction of development, because new technologies allow the dispersion of heat sources based on renewable energy. Therefore, new buildings would not have to be located densely. However, research conducted in Denmark [124] proves that it is profitable to use traditionally designed district heating networks in conjunction with low-temperature systems based on renewable energy sources. Danish researchers even emphasise the competitiveness of such solutions with a dispersed system of ground heat pumps.

A critical opinion on the lack of coordination of the spatial development of cities with the existing heating network or other existing networks of technical infrastructure results from the observation of growing, non-internalised development costs and extensive incorporation of space for sprawling cities. In this context, Polish researchers ask the question: is a smart city a utopia or is there a chance to implement solutions that fulfil ambitious goals, such as innovative use of technology, efficient transport, efficient use of energy and a clean environment [125]? The answer indicates that developing a strategy for smart cities is a necessity from the point of view of future generations. The conclusions of researchers assessing the condition of heating systems in the Baltic Sea basin are consistent with the conclusions of the Danes [126]. Increasing the efficiency of heating networks and their gradual transformation require the replacement of heat sources and their decentralisation; however, it should also take into account the use of the potential of the existing network for heat supply. Hence, there is a conclusion regarding the necessity to coordinate the spatial development of the settlement structure with the existing network infrastructure. Taking up the topic of extensive urban development also refers to the existing acts constituting the framework of development policy, such as the Paris Agreement [127] or the European Green Deal [128]. The abovementioned considerations and research results encourage in-depth reflection on all the costs of spatial development, both from the political and scientific perspective. There is also the third important group of stakeholders, developers, who should not only corporate socially responsible, but also corporate environment responsible [129]. There is still lack of data that could provide unambiguous arguments confirming the negative effects of urban sprawl, but unless scientific proved it is still the myth for most of the regular households. To support responsible land use policy, it is necessary to launch scientific research aimed at developing models for assessing the costs of land use policy, not only in Poland, but also in the comprehensive manner with international context.

## 5. Conclusions

The article contributes to the discussion on the costs of spatial chaos and challenges for land use policy. This discussion is gaining importance around the world as it relates to the processes of land consumption and the development paradigm seen as an increase in production and in consumption, in general. The aim of the article was, on one hand, to introduce the problem and present it to a wider group of readers, and on the other hand, to recommend changes in the conduct of spatial policy by reducing the non-internalised costs. Conclusions from the discussion presented in the article are of an application character.

As a result of the lack of the spatial policy coordination in functional areas, suburbs are still developing vigorously, which entails costs related to the expansion of dispersed technical infrastructure. Meanwhile, the potential of city centres with water supply, sewage systems and heating networks is not used. It happens as a result of incomplete economic calculation and non-internalisation of social and environmental costs as well as operating costs of developing scattered housing areas. An additional problem is the lack of concentration of revitalisation activities in cities and their dispersion over a vast area of intervention, which does not allow a significant and convincing change. It is necessary to rationalise land management policies, understood as the coordination of spatial development with, inter alia, energy efficiency policy and the development of technical network infrastructure, in particular using the potential of the existing heating network.

Research on the effectiveness of spatial development must be extended to include non-internalised costs as well as the analysis of alternative solutions using, for example, the revitalisation of urban areas with already developed sewage, water supply or heating systems. This thread is still poorly analysed in scientific articles. The presented paper presents for the first time this context in relation to spatial development processes in Poland.

**Author Contributions:** Conceptualization, methodology, software, validation, formal analysis, investigation, resources: A.J.-S. and M.Z., writing—review and editing—A.J.-S. and M.Z. All authors have read and agreed to the published version of the manuscript.

**Funding:** This research received no external funding.

**Institutional Review Board Statement:** Not applicable.

**Informed Consent Statement:** Not applicable.

**Data Availability Statement:** The data presented in this study are available on request from the corresponding author.

**Acknowledgments:** The authors would like to thank Edyta Tomczyk for her help with the translation and language consultations.

**Conflicts of Interest:** The authors declare no conflict of interest.

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
