# Peer review of "Alternative between Revitalisation of City Centres and the Rising Costs of Extensive Land Use from a Polish Perspective"

_land, doi:10.3390/land10050488_

Round 1
Reviewer 1 Report
The paper aims at interpreting the effects of the urban sprawl trends in Poland also highlighting policies, governance and management models oriented at encouraging land-consumption behaviours. This topic has been extensively studied in the last decades in different geo-political contexts, the paper needs to be updated with the most relevant literature in the field.
The theoretical background is poor and the references to Naples and Copenhagen seem to be out of context, inconsistent with the paper rationale and not adequately supported by relevant literature.
The methodology is outdated and the focus is limited respect the aims of the papers. The results need to be better presented and linked to the general aims of the paper and to the discussion of the case study.
The topic could be interesting if adequately related to the massive literature in the field at international level.
The paper could benefit from an extensive conceptual reorganization of the contents and a coherent presentation of the results, according to the abstract and introduction.
Author Response
|
1.1. The paper aims at interpreting the effects of the urban sprawl trends in Poland also highlighting policies, governance and management models oriented at encouraging land-consumption behaviours. This topic has been extensively studied in the last decades in different geo-political contexts, the paper needs to be updated with the most relevant literature in the field.
The topic could be interesting if adequately related to the massive literature in the field at international level. |
Additional analysis of the international literature was carried out. The introductory part of the article has been supplemented. The discussion was also expanded to include the international aspect and conclusions resulting from the published research.
|
|
1.2. The theoretical background is poor and the references to Naples and Copenhagen seem to be out of context, inconsistent with the paper rationale and not adequately supported by relevant literature. |
The theoretical background has been amended with arguments and quoted literature. References to Naples and Copenhagen are intended to indicate the relationship between the declining and in crisis city cores and the process of suburbanization. That is why the process of comprehensive urban revitalization is one of the key instruments to counteract the cost-intensive and extensive development of suburbs in the functional areas of cities. |
|
1.3. The methodology is outdated and the focus is limited respect the aims of the papers. The results need to be better presented and linked to the general aims of the paper and to the discussion of the case study. |
The description of the methodology has been improved. Modern and currently relevant research methods such as statistical analysis and analysis with the use of GIS tools were indicated. The results were linked to the overall objectives of the article based on the cited cases. |
|
1.4. The paper could benefit from an extensive conceptual reorganization of the contents and a coherent presentation of the results, according to the abstract and introduction. |
The paper was conceptual reorganised and amended massively. |
Reviewer 2 Report
The submitted manuscript is interesting and original with reference to the issue of analyzing and assessing urban sprawl phenomena. The study defines and implement a methodological approach to charatcetrize the patterns of the Polish spatial structure of public infrastructure, namely sewage (with reference to the whole Polish Country) and heating (as regards the Dolnośląskie Voivodeship) systems, and of the implementation of planning policies aimed at revitalizing degraded areas indentified by the local municipalities.
The research framework is based on the following conceptual points (p. 4):
i) trends of Polish local governments: sanctioning the effects of urban sprawl in peripheral areas by equipping them with technical infrastructure; planning revitalisation activities in degraded and revitalisation areas that are vast and numerous in terms of population;
ii) spatial management implemented by the local municipalities in order to compete for attracting residents by: creating conditions for locating buildings on the largest possible area of the municipality, and, at the same time, by avoiding restrictions on the development of areas that are not equipped with utilities (e.g. sewage system or district heating network); trying to enable external subsidies for revitalisation activities on as large area as possible;
iii) research objectives of the submitted manuscript, on the basis of the previous premises: presentation of the scale of changes in equipping municipalities with technical infrastructure on the example of a sewage network; presentation of the Polish approach to designating vast revitalisation areas with numerous inhabitants; presentation of the discrepancy between the district heating network routes in municipalities and the directions of building developments in them.
A detailed assessment of the three research objectives listed above is implemented, and the results are clearly presented in the manuscript
In my opinion, the study should not be published in its present form, since a number of caveats need to be addressed in a proper way.
In a revised version of the study, the authors should carefully address the following points.
i. Section 1. “Introduction.” Since the study aims at characterizing the Polish planning approach to the research objectives reported in point iii) above, a thorough literature review should be added in order to make the reader aware of how the scale of changes in equipping municipalities with technical infrastructure (e.g. the sewage or the heating networks) has played a distinguished role in managing revitalization policies and related measures to address the impacts of urban sprawl. Specific alternative approaches adopted in other countries should be associated and compared to the Polish planning policy framework. This is a fundamental issue which would make the submitted article interesting and appealing to the readers of Land.
ii. After Section 4, I would recommend the authors add a “Discussion,” since a discussion is really missing. In my view, the authors should present a detailed analysis of the results in the light of the studies, available in the international literature, concerning the analysis of the organization of urban sewage and heating networks and their relations to revitalization measures. Furthermore, I would recommend the authors analytically discuss the advancements implied by their manuscript as compared to the current literature, in order to make the reader aware of the value added of the submitted article.
iii. After the (hopefully) added Section 5, I would recommend the authors add a “Conclusion,” since a conclusion is really missing as well. I would recommend the authors discuss the implications of the outcomes of the study as regards planning policies concerning urban sewage and heating networks and their relations to revitalization measures. A profile that needs to be addressed concerns the analogies and differences which characterize, in terms of planning policies, either at the national or at the Voivodeship level, the municipal categories identified in the manuscript, namely: cities with a county status; urban municipalities; urban-rural municipalities; rural municipalities.
Moreover, I would recommend the authors address the issue of the exportability of their study to international contexts different from the Polish framework, in order to make the reader aware of the reasons the submitted manuscript is likely to be helpful in addressing the assessment of the formation and development of urban planning decision-making processes with reference to other countries’ spatial contexts.
iv. A number of typos are identified in the text, such as “leasure” (line 821), so I would suggests the authors revise the English of the manuscript.
Author Response
|
Section 1. “Introduction.” Since the study aims at characterizing the Polish planning approach to the research objectives reported in point iii) above, a thorough literature review should be added in order to make the reader aware of how the scale of changes in equipping municipalities with technical infrastructure (e.g. the sewage or the heating networks) has played a distinguished role in managing revitalization policies and related measures to address the impacts of urban sprawl. Specific alternative approaches adopted in other countries should be associated and compared to the Polish planning policy framework. This is a fundamental issue which would make the submitted article interesting and appealing to the readers of Land. |
The literature was supplemented. The approach represented by the authors is a novelty, in particular with regard to the situation of spatial planning and revitalization processes in Poland. The analysed literature allows for the conclusion that the research direction referring to the search for the least costly ways of spatial management is correct. The article proves that in Polish conditions there is no coordinated approach to the management of functional areas of cities. The cost related to the dispersed development of technical infrastructure is not a sufficient argument to limit extensive development. Moreover, the aforementioned lack of coordination of development policy in functional areas makes decisions on the development of new housing estates independent of the revitalization of areas equipped with technical infrastructure. It is also a contribution to the debate on spatial development in the future, also in the context of the green deal. |
|
After Section 4, I would recommend the authors add a “Discussion,” since a discussion is really missing. In my view, the authors should present a detailed analysis of the results in the light of the studies, available in the international literature, concerning the analysis of the organization of urban sewage and heating networks and their relations to revitalization measures. Furthermore, I would recommend the authors analytically discuss the advancements implied by their manuscript as compared to the current literature, in order to make the reader aware of the value added of the submitted article.
|
Part of the discussion has been supplemented. As suggested, a detailed analysis of the results was presented in the light of the studies available in the international literature concerning the analysis of the organization of municipal technical infrastructure networks. A new approach was highlighted. |
|
After the (hopefully) added Section 5, I would recommend the authors add a “Conclusion,” since a conclusion is really missing as well. I would recommend the authors discuss the implications of the outcomes of the study as regards planning policies concerning urban sewage and heating networks and their relations to revitalization measures. A profile that needs to be addressed concerns the analogies and differences which characterize, in terms of planning policies, either at the national or at the Voivodeship level, the municipal categories identified in the manuscript, namely: cities with a county status; urban municipalities; urban-rural municipalities; rural municipalities. Moreover, I would recommend the authors address the issue of the exportability of their study to international contexts different from the Polish framework, in order to make the reader aware of the reasons the submitted manuscript is likely to be helpful in addressing the assessment of the formation and development of urban planning decision-making processes with reference to other countries’ spatial contexts.
|
A section "conclusions" has been added, in which the most important conclusions from the discussion and the research confrontation were synthetically formulated. The improved narrative and vocabulary used will certainly allow you to better understand the intention of the authors. The added context of the research results presented in international literature makes the text more universal. |
|
A number of typos are identified in the text, such as “leasure” (line 821), so I would suggests the authors revise the English of the manuscript.
|
The article was proofread. |
Reviewer 3 Report
This is an interesting manuscript and focuses on a topic of sure interest for the readership of Land journal. However, it has some unclear issues. Please see the following list of comments and suggestions:
Comments and suggestions:
- The title should be more specific.
-The abstract should be sharper and more critical.
- The introduction could be clearer, explain more the aims and purpose. You should engage more with the relevant academic literature. It includes few sources for literature review part. An extensive literature review has to be included to understand what has been researching in the field and help to identify research gaps for your research.
- What are the innovative contributions of your manuscript to science? (this needs to be clearer in your manuscript).
- Please include a methodological framework in the methods and material section.
- The methodology needs more explanations regarding alternative approaches.
- The discussion is absent. You need to include an in-depth discussion on those important results including compare your results with previous studies and explain why your results are similar or differences with previous findings.
- Can the authors suggest how these results can be used in the decision-making process? please emphasise the contribution and implication of your manuscript.
- Minor grammar and punctuation errors can be found throughout the text and need to be corrected.
Author Response
|
The title should be more specific.
|
The title has been corrected as suggested. |
|
The abstract should be sharper and more critical. |
The abstract has been shortened and is more critical. |
|
The introduction could be clearer, explain more the aims and purpose. You should engage more with the relevant academic literature. It includes few sources for literature review part. An extensive literature review has to be included to understand what has been researching in the field and help to identify research gaps for your research.
|
The introduction has been expanded and structured, which clearly indicates the objectives of the article indicating the need for an in-depth analysis of the costs of spatial development and the internalisation of environmental and social costs. The introduction contains more references to the relevant literature, showing an overview of scientific achievements in the context of the costs of spatial chaos, including spatial planning in connection with energy efficiency. |
|
What are the innovative contributions of your manuscript to science? (this needs to be clearer in your manuscript). |
The manuscript provided the answer about the innovative contribution to science. Scientific evidence should be the source of changes in the pursued development policy. Research on the effectiveness of spatial development must be extended to include non-internalized costs into the calculation of spatial chaos costs. as well as the analysis of alternative solutions using, for example, the revitalization of urban areas with an already developed sewage system, water supply or heating. The article concentrates on these three symptoms of spatial chaos in Poland: random and dispersed expansion of the new investment in a sewage system, lack of integration between district heating systems and direction of residential development and dispersed effects of revitalisation, which cannot overwhelm flight from blight. The obtained results allowed to confirm the thesis about the extensive land use policy model in Poland. This thread is poorly illustrated in scientific articles. The presented article presents this context for the first time in relation to the processes of land use policy in Poland. |
|
Please include a methodological framework in the methods and material section. The methodology needs more explanations regarding alternative approaches.
|
The description of the methodological framework has been improved and made clearer. The broadcasting approach based on statistical analyses and the use of GIS tools is explained. |
|
The discussion is absent. You need to include an in-depth discussion on those important results including compare your results with previous studies and explain why your results are similar or differences with previous findings |
The part of the article devoted to the discussion was expanded and juxtaposed to the available research results of other authors. The problem studied and presented by us was noticed in other countries (in Europe and North America). Our article contributes with observations from Poland to a broad scope of literature cited. |
|
Can the authors suggest how these results can be used in the decision-making process? please emphasise the contribution and implication of your manuscript.
|
The article contributes to the discussion on the costs of the extensive land use policy in Poland. This discussion is gaining importance around the world as it relates to the processes of space consumption and the development paradigm seen as an increase in production and an increase in consumption in general. The aim of the article is to present three phenomena (random and dispersed expansion of the new investment in sewage system, lack of integration between district heating systems and direction of residential development and dispersed effects of revitalisation) and to recommend changes in the land use policy by reducing the non-internalized costs. This thread was expanded in discussion and shortly summarised in the added conclusion. |
|
Minor grammar and punctuation errors can be found throughout the text and need to be corrected. |
The article has been proofread. |
Round 2
Reviewer 1 Report
The paper has been considerably improved although it could benefit from further review of the sections dedicated to the results and the discussion to make the text more readable for an international audience
Author Response
Dear Sir or Madam,
thank you for the previous comments that helped us improve the article and complete the international research context. We thought about the extent to which we should further strengthen it and decided to add a few summary general recommendations at the end of the Discussion. We removed a short paragraph in the Methodology, that we found too local and understandable mainly for the Polish reader.
We hope that the article - although it still mainly presents a Polish perspective - may also be a source of inspiration for similar research in other countries.
The article has been carefully proofread and all corrections have been marked in the text in accordance with the instructions from the Land Editorial Office.
Reviewer 2 Report
All the points I addressed in the first place are appropriately addressed in the revised version of the study. As a consequence, I would suggest Land accept the resubmitted manuscript in its present form.
Author Response

(The authors gave the same response as above.)

Reviewer 3 Report
Thank you and your colleagues for the modifications that you have made to this article and how well you have responded to the suggestions.
Author Response

(The authors gave the same response as above.)
